# Digital health interventions for improving access to primary care in India: A scoping review

Lenny Vasanthan[1,2], Sindhu Kulandaipalayam Natarajan[3], Andrew Babu[1], Mohan S. Kamath[4], Sureshkumar Kamalakannan[5,6]*

1 Department of Physical Medicine and Rehabilitation, Physiotherapy Unit, Christian Medical College, Vellore, Tamil Nadu, India, 2 Department of Physiotherapy, Honorary Clinical Senior Lecturer, Melbourne School of Health Sciences, University of Melbourne, Melbourne, Australia, 3 Division of Gastrointestinal Sciences, Wellcome Trust Research Laboratory, Christian Medical College, Vellore, Tamil Nadu, India, 4 Department of Reproductive Medicine and Surgery, Christian Medical College, Vellore, Tamil Nadu, India, 5 Department of Social Work, Education, and Community Well-being, Northumbria University, Newcastle Upon Tyne, United Kingdom, 6 Institute of Public Health Sciences, Public Health Foundation of India, Hyderabad, India

☯ LV and SKN share the first authorship
* sureshkumar.kamalakannan@northumbria.ac.uk

**Data Availability Statement:** All data and related metadata underlying the findings is reported and available within this manuscript.

## Abstract

Access to quality healthcare remains a challenge in low-and middle-income countries. Vulnerable populations with unmet needs face the greatest challenge in accessing primary care for appropriate and timely healthcare. The use of digital technologies can not only strengthen health systems but also improve access to health care, particularly for the vulnerable. This scoping review aims to assess the various digital health technologies and interventions available for improving access to primary care for the vulnerable in India. This scoping review employed the Joanna Brigg Institute's (JBI) guidelines and Arksey and O'Malley's methodological framework. The literature search was conducted in Medline/PubMed, Embase, Web of Science—Core Collection, Scopus, AgeLine, PsycINFO, CINAHL, ERIC, Cochrane CENTRAL, and Cochrane Effective Practice and Organisation of Care (EPOC) Group Specialised Register databases, using the keywords, such as 'Access', 'Healthcare', 'Assistive technology', 'Digital health' 'Vulnerable', 'India' and 'Healthcare technology'. A two-staged screening of titles and abstracts, followed by full-text was conducted independently by two reviewers, using the Rayyan software. Subsequently, the data was extracted from selected studies using a pre-designed and approved extraction form. The data was then synthesised and analysed narratively. The protocol for this review has been registered with open science forum (OSF) registries (https://osf.io/63pjw/). The search yielded about 3840 records, 3544 records were eligible for screening of titles and abstracts. We included seven studies after a two-round screening and identified seven different technological innovations developed to bridge gaps in access to primary care. The commonly used digital health interventions for improving access to primary care were virtual tele-health systems and mHealth applications in-built within an android smartphone or a tablet. Digital health interventions was either used as a standalone tele-health aid or a collaborative system for community workers, primary care physicians as well as the health service users.

**Funding:** This review was funded and written as part of the research for the Lancet Citizens' Commission on Reimagining India's Health System. The Lancet Commission has received financial support from the Lakshmi Mittal and Family South Asia Institute, Harvard University; Christian Medical College (CMC), Vellore; Azim Premji Foundation, Infosys; Kirloskar Systems Ltd.; Mahindra & Mahindra Ltd.; Rohini Nilekani Philanthropies; and Serum Institute of India. The views expressed are those of the author(s) and not necessarily those of the Lancet Citizens' Commission or its partners. The study was supported by CSR funding from Infosys Limited, Bangalore. None of the authors listed in this review were specific recipients of the fund. Grant number: 22G191.

**Competing interests:** The authors have declared that no competing interests exist.

The purpose of these innovations was to increase awareness and knowledge to access support for specific aspects of healthcare. Virtual primary health care with the specialist in the hub supporting general physicians at the primary health centres in blocks and districts was another such model used for improving access to primary care. Digital health interventions was also used for mass community screening of disabilities, such as persons with hearing disability. To re-imagine a digitally empowered health systems in India, also inclusive of the vulnerable, it is important to inclusively conceptualise, systematically develop and rigorously evaluate any public health interventions including those that are enabled by digital health interventions to bridge the gaps in access to primary care in India. Such a strategy could address the paucity of evidence in public health interventions and provide sustainable strategies to strengthen health systems in India.

**Trial registration: Open Science Framework—Registration Link**: https://osf.io/63pjw/.

## Introduction

Health care is a vital commodity that is either provided free of cost by the state or public sector, or at a cost that may or may not be subsidized by private sectors, civil societies or their combined partnerships in India [1]. According to the National Family Health Survey (NFHS-5) done between 2019 and 2021, the public health sector is the main provider of healthcare for 52% of households in urban areas and 47% in rural areas, while the private sector is the primary provider of healthcare for 52% of people in urban areas and 46% in rural areas [2]. Right to healthcare for all Indians is a priority for the Government of India [3]. However, equity of access to healthcare irrespective of socio-economic status, caste, region, or income is yet to be achieved uniformly across all Indian settings [4]. Challenges remain in large with reference to optimal utilization of healthcare provided by the government sector including poor quality of care (48%), geographical inaccessibility to government healthcare facilities (45%), and long waiting times to seek consultation and treatment (41%) [5]. Also, the healthcare system in the country is financially supported with only a meagre 1.26% of the total Gross Developmental Product (GDP), which subsequently harbingers the private sector stepping in to meet the rising health demands, implying an increased out-of-pocket expenditure from the consumers, especially impacting the people in lower socio-economic strata [6]. The barriers to effective utilization and poor access to health care in India need to be addressed meticulously and holistically to achieve universal health coverage (UHC) [7].

Inequity in the utilisation of healthcare services exist in India and this may be attributed to reasons such as the caste hierarchy, social status, opportunity costs, distance to the nearest health facility, convenience, road or transport conditions, etc [8]. An important factor for inequitable utilisation of primary care is inadequate availability of comprehensive healthcare services for those especially with non-communicable diseases as a result of epidemiological transitions over the past three decades in India [9]. When the above factors are crucial in the access of healthcare by the general population, the access is further challenging if the person is vulnerable [10]. The vulnerable include those who are economically backward, elderly, living in rural regions of India, and individuals who are differently abled, experience twice the burden of being ill and also have poor access to health care in India [11]. These vulnerable population are at a substantial disadvantage to even access basic health care in India. Vulnerability and inaccessibility to healthcare constitute a vicious cycle exposing the vulnerable to the

bottom most strata of the neglected with reference to healthcare access needs. Though several efforts have been taken to address these issues, the situation has remained unchanged for decades [12].

One of the strategies that could potentially be implemented to improve access to healthcare in India is embracing the tremendous advances in technology [13]. This, especially in the case of vulnerable populations, as an extremely helpful aide. digital health interventions (DHI) also called as assistive technologies (AT) in health are known to enhance efficiency in health systems worldwide [14]. DHI generally include any device, product, or service that could help persons with any kind of health issues or disability to carry out their activities of daily living that are limited in varying degrees by their illness and/or disability [15]. However, for the purpose of this scoping review, we would like to define digital health interventions as a strategy that enhances access to primary health care for people with any health condition that requires attention from healthcare providers. Even globally, there is an unmet need for DHI specifically for people with disabilities who are in dire need of them [16]. The main reason is that access to DHI in the form of assistive or digital devices at the primary care level is almost non-existent in the national health systems, in many low and middle-income countries (LMIC) including India [17].

Digital health interventions are being increasingly adopted and being provided for those in need by healthcare providers through the existing and the healthcare levels and systems in India [18]. However, the challenge faced currently is in enhancing the efficiency of the systems through which providers of DHI deliver care to those in need rather than addressing the gaps in barriers to accessing primary care [18]. There is not much information available on factors that influence this strategy for effective, safe and good quality primary care delivery using DHI in the country [19]. It is therefore highly important to understand the approaches in utilisation of DHI for primary care, especially from the perspective of enhancing access to health care in India. Considering the glaring gap in supply and demand in the access to DHI in healthcare, it is important to study various modalities of DHI that could be used and potentially scale up, to bridge the gap in access to primary care, and hence this scoping review was conducted to study the available literature on the use of DHI in enhancing access to primary care in India.

The objective of this scoping review is to identify the available DHI that can empower the vulnerable to access primary care in India. Access to primary care is important as it provides an opportunity to maintain or improve one's health through the existing health systems without encountering any catastrophic expenditure. Given the very recent COVID-19 pandemic situation where access to healthcare became very limited and the demand for healthcare needs increased tremendously (inverse care law), it is crucial to gather in-depth insights on how digital health interventions in the form of digital health interventions may improve or provide access to primary care in India. This would help prepare the vulnerable to access the needed information on handling similar situations in the future. This will also enable systems and primary care provision of safe, effective, and good quality, yet affordable healthcare. This scoping review will help sensitise the health and policy makers to identify the need for improving access to health services especially to the vulnerable population, thereby improving health equity in India.

## What is Unique about this scoping review?

In health and social care literature, digital health interventions are majorly described either from a technological perspectives or from a disease prevention, treatment or management perspectives. Very hardly we find literature on the use of digital health intervention that focuses

on improving access to primary care especially for people who are most vulnerable. This scoping review aims to review the literature about this niche aspect.

## Methods

### Design of the review

Digital health interventions for improving access to primary care in India is a highly diverse and heterogeneous area which could include various modalities and levels of primary care, and hence, needs to be explored multi-dimensionally and holistically. Harnessing the advantage offered by a scoping review approach for such a broad area, we conducted this review based on Arksey and O'Malley's framework, along with the subsequent improvements of the framework by the Joanna Briggs Institute (JBI) and the Preferred Reporting Items for Systematic reviews and Meta-Analyses extension for Scoping Reviews (PRISMA-ScR) checklist [20–24]. In order to avoid duplication of the work being undertaken and to provide accessibility of the protocol widely, the protocol of this scoping review was registered with the Open Science Framework (OSF) [25].

### Identifying research question

A search was conducted on the currently available literature and internal discussions were held with experts in the field to frame the overarching research question: "Can digital health interventions for the vulnerable population improve access to primary care in India".

### Identifying relevant studies

We searched relevant, peer-reviewed and published studies from Medline/PubMed, Embase, Web of Science—Core Collection, Scopus, AgeLine, PsycINFO, CINAHL, ERIC, Cochrane CENTRAL, and Cochrane Effective Practice and Organisation of Care (EPOC) Group Specialised Register databases. The search was first run-on the 29[th] of November 2022 and was an updated search was run 6 months later on 30 April 2023. Given that this scoping review is specific to the Indian setting, publications were expected to be predominantly from Indian journals that may or may not be indexed in international electronic databases and hence, databases specific to the Indian context such as Ind-med, and National informatics were additionally included.

Three key concepts 1. Access to primary care; 2. Vulnerable population; and 3. digital health technologies / assistive technologies were used to build the key terms for the primary search strategy for this scoping review. Hand search and grey literature search were also performed with combinations of the above key terms, and this was done to include relevant literature from official reports, guidelines, advice, and recommendations (e.g. from national or international agencies, non-governmental organizations, or public health authorities). Finally, we also consulted key stakeholders and experts in public health and requested for additional references that may not still have been included following the above describes search strategy.

### Study selection and sources for searching evidence

All types of empirical studies, from the community as well as hospital settings in India were considered. The primary eligibility criteria for the studies to be included in this review were.

1. **Access to primary care**: Studies that provided clear information on access to primary care in India.

2. **Digital health interventions**: Studies related to DHI primarily aimed at bridging the gaps in inaccessibility related to primary care in India.

**3. Vulnerable Population**: Studies that focussed on access to primary care by the vulnerable population as defined in this study protocol.

All studies published on this research question to date were searched and search results were uploaded in an open access tool "Rayyan" which is used for screening and appraising studies related to systematic reviews [26]. They were searched and reviewed by two independent reviewers. A consensus was arrived by the scoping review team that pre-prints will not be included in this scoping review, as pre-prints are not peer reviewed and can impact the final outcomes if these studies change results/conclusions at a later stage. Only literature published in English were selected for this review. The selection of studies was through the three-step method as recommended by JBI.

Further, studies that have also used a comparator or a control arm where either no digital health interventions were used or different types of digital health technology were compared were also included in the review. The final outcome focussed on the development, application, and changes/differences in the use of digital health interventions to bridge the gaps in the access to primary care by the vulnerable population in the Indian setting at various levels (urban/rural/tribal; low/middle/high socio-economic status; male/female/ children; and disabled/abled;).

## Charting the data

Two reviewers (LV and KNS) piloted the data extraction form developed by the research team with formal data elements that included publication type, and source to extract the data. The reviewers completed a pilot extraction of the data with a random sample of 5% of the included studies which was verified by a third reviewer (SK). The abstract-and-titles screenings and the full-text assessments were made against the eligibility criteria and were conducted by two independent reviewers (LV and KNS.), after pilot screenings with over 80% agreements, overseen by the leading review author (SK). Any discrepancies were resolved through consensus or the leading author's input. We followed a predetermined coding structure based on the pilot exercise to extract and chart the data from the included studies. The data from the included studies was extracted by two reviewers independently (LV and KNS). The two reviewers extracted text quotations on access to primary care, with a specific focus on vulnerable populations as well as innovative DHI. Any DHI and related programmes or interventions developed, as well as further recommended, that promoted equitable access to primary care in India were also extracted.

## Collating, summarizing, and reporting the results

This scoping reviews aims to provide a summative description of the amount and range of the related literature on DHI for the vulnerable in improving access to primary care in India. Descriptive statistics were used (e.g., percentages) to study the type of publication type, region where the study was done i.e. state (or states) addressed, the database source (e.g., databases of peer-reviewed literature, or Google searches on the grey literature), and different type of vulnerable population (poor, disabled, children, elderly), purpose of DHI (for decision making, for patients, or for the system) and issues in relation to affordability, availability, appropriateness, accessibility, and approach, when applicable.

The analyses for the review were derived from an initial, deductive coding, that is, based on a predefined coding structure built by the research team, performed independently by the two data extractors (LV and KNS), along with any supporting qualitative notes or text quotations. These supportive notes were enabled the scrutiny of the remaining elements. Final decisions on any disagreement in the ratings were made by the guarantor of the review, who led the

design but had no primary reviewer roles (MK). Finally, a qualitative thematic analysis was conducted from the content (i.e., text quotations) extracted from the literature.

### Experts' consultation

The consultation of experts was an optional yet recommended step in this scoping review with an objective to find additional relevant publications and to seek reinterpreting of the review results and to further understand finer implications by deductive reasoning. Experts for both steps were identified and consulted. Regarding the finding of relevant publications, as mentioned previously, experts were identified and supplied with a preliminary list of inclusions and consulted as key informants on any additional reference potentially fitting the inclusion criteria that may have been missed. Although this process might not ensure exhaustive coverage of the grey literature, we believe that it may contribute to closing gaps in the representativeness of the reviewed information. Finally, the same group of experts was provided with the opportunity to comment or suggest amendments to the first complete draft of the results and discussion.

## Results

The search yielded a total of 3840 records. After removing duplicates and ineligible records that are ineligible, 29 studies were identified for full-text screening. Following full-text screening of these 29 studies, seven studies were eligible for inclusion in this review. Details of the search and the process of identification of studies for inclusion in the review are provided as a PRISMA flow diagram in Fig 1. We expected significant heterogeneity among the included studies and hence did not conduct any meta-analysis, but conducted a narrative synthesis.

### Characteristics of included studies

Overall, seven studies were included in the review [27–33]. Of these seven studies, two studies were from Uttar Pradesh [27–29], two studies were from Karnataka [30–32], one each from Bihar [28], and Jharkhand [33] respectively, and another one included data from 12 Indian States [31]. All identified studies were published in the past 5–6 years. Five studies were conducted at the level of Primary Health Centre (PHC) [27–29,32,33] and two studies were conducted within the community where the participants lived [30,31]. Pregnant women, mother, children, homeless people with mentally illnesses, people addicted to drugs, people living in remote, hard to reach rural regions and the under-privileged and under-served from urban slums were the vulnerable population who participated in the included studies. More details about the included studies are provided in Table 1.

### Digital health interventions for access to primary care

Although we identified about 262 records during the first level of screening process and 29 records during the second level screening process, only seven included studies described the concept of access to primary care using digital health interventions [27–33]. The other studies were based on use of DHI but not with reference to primary care access or for the vulnerable population. The commonly used DHI for improving access to primary care were virtual tele-health systems and mHealth applications in-built within a smartphone or a tablet. digital health interventions were conceptualised and used for various purposes among the included studies. It was used as a standalone tele-aid for people to increase awareness and knowledge on healthcare access and support for various disease conditions [27,30,33]. Virtual primary health care with the specialist/s in the hub or higher referral centers, supporting general physicians at

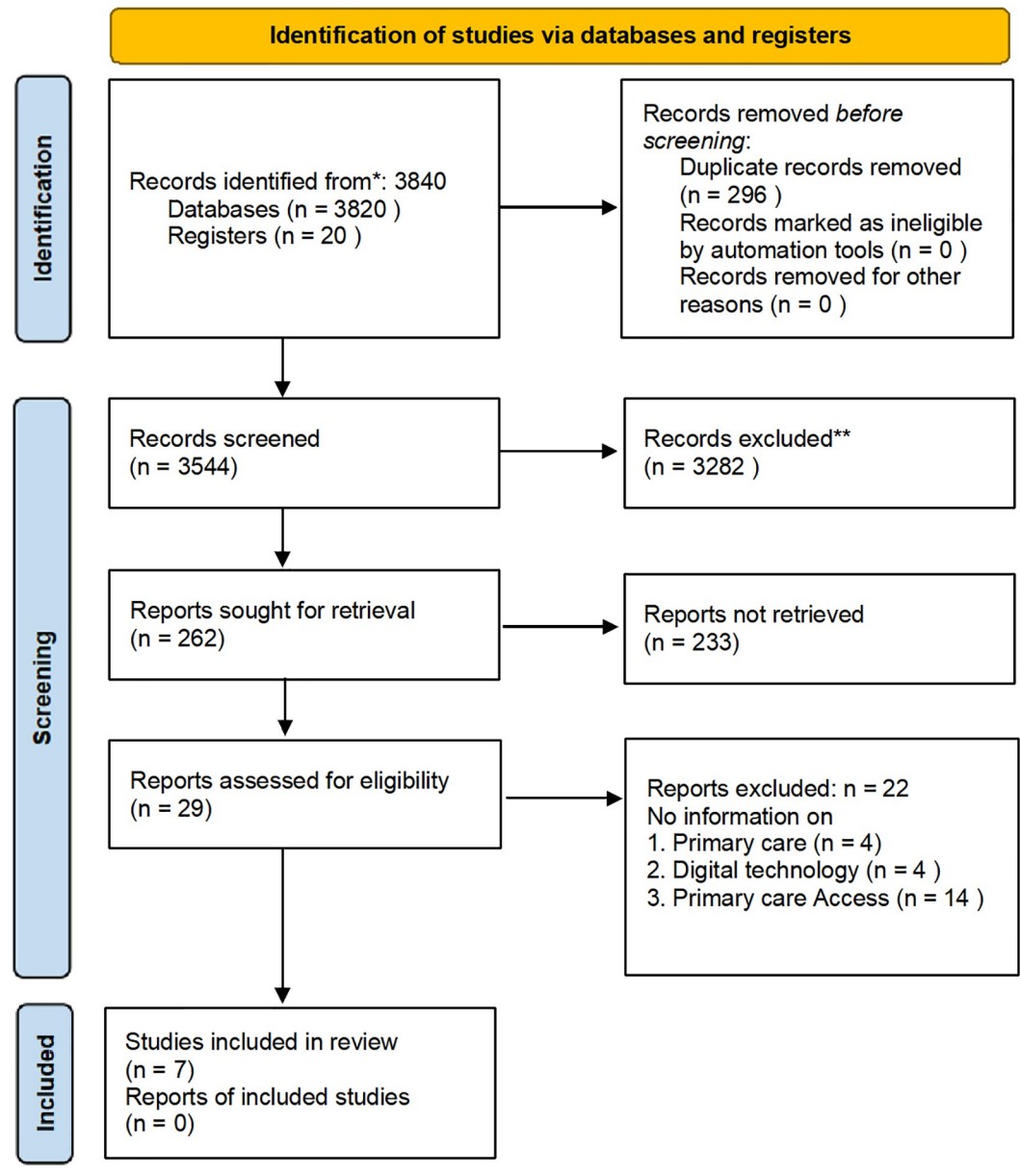

*Consider, if feasible to do so, reporting the number of records identified from each database or register searched (rather than the total number across all databases/registers).

**If automation tools were used, indicate how many records were excluded by a human and how many were excluded by automation tools.

*From:* Page MJ, McKenzie JE, Bossuyt PM, Boutron I, Hoffmann TC, Mulrow CD, et al. The PRISMA 2020 statement: an updated guideline for reporting systematic reviews. BMJ 2021;372:n71. doi: 10.1136/bmj.n71 For more information, visit: http://www.prisma-statement.org/

**Fig 1. Study selection process in the review.**

**Table 1. Key characteristics of the included studies (N = 7).**

| Study ID | State | Vulnerable Group | Level of Primary Care | Digital health interventions & Category | Aspect of Accessibility |
|---|---|---|---|---|---|
| Prinja 2017 [27] | Uttar Pradesh | Maternal, new-born and child health care services | PHC/Blocks | Tele-aid | Increasing awareness to access maternal and child health (MCH) services |
| Sagi 2018 [28] | Bihar | Drug Addiction | PHC | Telementoring | Access to treatment advice from Tertiary care specialists |
| Angrish 2020 [29] | Uttar Pradesh | Primary health care in rural and hard to reach areas | PHC | Virtual PHC | Coverage and access to primary care services |
| Garner 2020 [30] | Karnataka (Bengaluru) | Urban slum and 11 rural villages | Community | mHealth App | Increasing awareness and knowledge about Hypertension. |
| Gupta 2020 [31] | 12 Indian states | Underprivileged and underserved community of rural and urban slums | Community | Shruthi Tele-otology program | Community screening of patients with ear diseases |
| Kulkarni 2020 [32] | Karnataka (Bengaluru) | Homeless people with mental Illnesses | Urban PHC | Collaborative Tele-psychiatry | Access to specialist Psychiatric rehabilitation |
| Choudhury 2021 [33] | Jharkhand | Pregnant women | PHC and SC | Mobile-based maternal education | Increasing awareness of access to care during pregnancy |

the primary health level in blocks and districts was another model for enhancing and improving access to primary care [28,29,32].

Digital health interventions in the included studies were used for preventive aspects particularly in the provision of mother and child health services and ante-natal care for pregnant women [27,32]. It was also used to provide specialist psychiatric and de-addiction services for difficult to reach populations, especially in the remote rural regions of India [28,29,30,33]. It was also utilised for the purpose of community-based screening of hearing loss and also as a device for mentoring primary care physicians to deliver healthcare services at the grassroot level [28,31].

## Digital health interventions for primary care

All seven studies included in the review used different DHIs for promoting access to primary care. The details of these digital health interventions are below. Table 2 provides a snapshot of these DHIs from the seven included studies.

**Mobile for mothers.** Mobile for mothers is an application to increase awareness about accessing care during pregnancy [33]. The software development was conceptualised by two non-governmental organizations (NGO), Network for Enterprise Enhancement and Development Support (NEEDS), an Indian NGO, and Simavi, a Dutch NGO. The application was designed for operation on java-enabled android smartphones for users who have either not studied or studied upto to a minimum of primary school level. The app runs on a free and open-source platform with four key modules. They are (1) registration, (2) antenatal care, (3) intranatal care, and (4) postnatal care. It also contains an Interactive Voice Recording System (IVRS) enabled program to provide pregnancy, maternal and child care information through texts, photographs, and voice prompts in the user's native language (Hindi) to pregnant women and mothers. The intervention was led by Accredited Social Health Activists (ASHAs) working for the National Rural Health Mission (NRHM).

**ReMIND mHealth intervention.** ReMIND mHealth intervention is very similar to the mobile for mothers application aimed at improving awareness and access to maternal, new born and child health services in rural Uttar Pradesh [27]. The device was conceptualised by two NGOs from ReMIND project and was developed on a java-based android platform by Dimagi Inc. The device was also developed as a job aid for ASHAs.

**Table 2. Details of the digital health interventions used for promoting access to primary care from included studies.**

| Study ID | Name of the Technology | Purpose | Conceptualization | Development | Evaluation |
|---|---|---|---|---|---|
| Prinja 2017 [27] | mHealth Intervention for ReMIND (Reducing Maternal and New-born Deaths) Project | An m-health intervention delivered for improving counselling to pregnant women through ASHA workers, improving coverage of services and generate demand for MNCH services. | Economic mobile phones (Java based) developed by Dimagi Inc, and 2 NGOs. Purpose: To serve as a job aid for ASHAs | Not reported | Statistically significant increase in -Coverage of iron-folic acid supplementation (12.58%) -Self-reporting of complication during pregnancy (13.11%) and after delivery (19.6%). -Increase in coverage of ≥3 antenatal care visits (10.3%), tetanus toxoid vaccination (4.28%), full antenatal care (1.1%) and ambulance usage (2.06%) which were statistically not significant. No change in institutional deliveries as part of ReMiND project. |
| Sagi 2018 [28] | Tele-mentoring for drug addiction | Access to specialist treatment advice | Telemontoring model: Includes both synchronous and asynchronous components *Synchronous component*: NIMHANS academic centre hub & remote district level primary care physician (PCP) spokes, implemented using low cost multipoint video conference based tele-ECHO clinics, held fortnightly. Mode: Remote PCPs used internet enabled smartphones to connect patient case summaries with NIMHANS hub *Asynchronous component*: Mobile-based e- learning integrated into a course completion certificate | Not reported | Feasibility assessment: Out of the 21 tele-ECHO clinics, 11 were held till end of August 2017. (N = 38 PCPs) All PCPs virtually joined at least one tele ECHO clinics. Of the 38 participants, two participated in all tele-ECHO clinics. 18 (47.36%) attended ≥6 tele-ECHO sessions. 89.47% completed 3 e-learning assignments 80% used smartphone with 4G connection to join tele-ECHO clinics Significant change in knowledge gained over 1 month (3.00 ± 0.86, $P < 0.001$) and 3-month (3.16 ± 0.90, $P < 0.001$) assessments compared to baseline (1.77 ± 1.02) 10 reported (32.25%) reported improved confidence in managing substance use disorder. |
| Angrish 2020 [29] | Virtual PHC e-clinic: Jiyyo E-Mitra Clinic | To expand health outreach in rural and hard-to-reach areas of India and provide primary health care services by connecting local practitioners and health workers visiting patients with qualified allopathic doctors in city through video conferencing technologies. | E-mitra clinic: A social business model innovated by Jiyyo Innovations (Chandigarh based start up). Jiyyo e-mitra clinic—Aim to reach rural & interior parts of India, connecting local practitioners and health workers (rural areas) with qualified allopathic doctors in city through video call. Location: Uttar Pradesh—initially implemented in 2 centres, gradually increased to 20. Consultation charges kept minimal —'no profit and no loss' model. | Not reported | Sample size: 800 Completed: N = 157, completed entire sessions of consultations, acceptability of the model. Consultations: General Physicians, specialists, super specialists (Paediatrics, Orthopaedics, Dermatology, Urology, Nephrology, Cardiology, Neurosurgery) Background: Rural (100%) Education level: Middle school (10%), primary school (21%) Audit of prescriptions: Revealed diagnosis, offered treatment, referral for advanced or complex cases. Also indicates feasibility of VPC in rural India. |

*(Continued)*

**Table 2.** (*Continued*)

| Study ID | Name of the Technology | Purpose | Conceptualization | Development | Evaluation |
|---|---|---|---|---|---|
| Garner 2020 [30] | mHealth application (app) | To improve health literacy on hypertension among participants in limited resource settings in India | App: 3D educational video (culturally appropriate) on mobile tablet. 7-minute 3D interactive health education, with animation. Teaching was on hypertension prevention, diagnosis and disease management. Mobile tablet served as a data literacy collection tool, before and after watching the video. App stored data on the tablet while wi-fi was accessible. | Not reported | Sample size: 346 There was a statistically significant improvement in test scores among participants after use of mHealth app, p<0.0001 with a 95% CI (2.23,2.76) and 345 degrees of freedom |
| Gupta 2020 [31] | ENTraview | To screen community level hearing using screening | Device used: ENTraview (Medtronic, Inc), a camera-enabled android phone integrated with an otoscope, audiometry screening, and a rechargeable, battery-operated light source The digital camera captures image of tympanic membrane, stores in the smartphone, noise isolating handset—enables screening in semi noisy environment Shruti—First technology driven program with integrated comprehensive ear care for persons with ear problems and can perform screening, diagnosis, treatment, and hearing rehabilitation | Not reported | Total numbers screened: 810,746 Ear problems: 265,615 (33%) *Impacted wax*—151,067 (57%) *Chronic Suppurative Otitis Media*—46,792 (18%) *Diminished hearing*—27,875 (10%) *Foreign body, Otomycosis & others*—27,152 (19%) *Acute Otitis Media*—12,729 (5%) 8% reported for treatment and received treatment at reduced cost through Shruti program partners. |
| Kulkarni 2020 [32] | Tele-psychiatric care model, for inpatients. Teleconsultations for evaluation & follow up care with primary care doctor | Access to rehabilitation services for people with mental illnesses | This collaborative care model refers to teleconsultations between the psychiatrist and primary care physician (PCP), in-charge of the patient at the NPK. Psychiatrist—guide the PCP, for screening, diagnosis and management of the patient with mental illnesses. The telepsychiatric consultations occur in the form of a 'hub-and-spokes' model. Hub: Tele Medicine Centre, Department of Psychiatry, NIMHANS. Participants: Consulting psychiatrists, Community Psychiatry Unit. Spoke: Located at the NPK. Participants: PCP along with patient whom referral was sought for. Consultation duration: 10–15 minutes (assessment), 10 minutes (follow up) | Not reported | Sample size: N = 132 Males—99, Females—33 Mean Age: 43.8±12.1 years Mean duration of teleconsultation: 7.8±4.9 hours Mean duration between Consultations: 2.1±1 months. Common diagnosis (initial) Unspecified non-organic psychosis—63 patients (47.7%) Schizophrenia—40 (30.3%) Psychosis with mental retardation—16 (12.1%) Depression—5 (3.8%) Bipolar disorders—5 (3.8%) Mental retardation—3(2.3%) |

**Table 2.** (Continued)

| Study ID | Name of the Technology | Purpose | Conceptualization | Development | Evaluation |
|---|---|---|---|---|---|
| Choudhury 2021 [33] | MFM—Mobile for Mothers | To improve awareness about pregnancy and care | MFM software—conceptualised by NEEDS, an Indian NGO and Simavi, a Dutch NGO. Designed for users with low-literacy, using Java or Android based smartphones, which run free and open source applications. MFM consisted of 4 modules 1) Registration, 2) Antenatal care 3) Intranatal care 4) Postnatal care Interactive Voice Recording System enabled mHealth provided maternal health information through texts, photographs, and voice prompts to pregnant women and mothers in Hindi language | Not-Reported | Quasi Controlled Intervention—Post intervention, awareness about tetanus toxoid injections and consumption of iron tablets improved significantly ($P < .001$) improved in the intervention group by 55% and 58%. Awareness about hygiene significantly ($P < .001$) increased by 57.1%. |

**Tele-mentoring for remote drug addiction management.** A tele-mentoring application was developed and assessed for its feasibility to be used by remote PCPs for the management of drug addiction in Bihar [28]. The innovation has both synchronous and asynchronous component. The National Institute of Mental Health and Neurosciences (NIMHANS) in Karnataka, one of the pioneering institutions for mental health in the country was conceptualised as the hub and the PCPs at the remote district level PHCs were the spokes. The tele-mentoring units at the spokes used a low-cost multi-point video conference facility based clinics called the Extension of Community Health Care Outcomes (ECHO). Internet-enabled smartphone were used by the physicians at the spokes to communicate with the hub equipped with a multi-disciplinary expert team to discuss cases and to manage patient care related to drug addiction. The physicians at the spokes were able to complete a mobile-based training and complete the course to be a part of this tele-mentoring solution that brought specialist care for drug addiction management at the PHC level asynchronously.

**Jiyyo e-mitra clinic.** This is an innovation to improve access to primary care to people living in hard-to-reach remote villages of Uttar Pradesh [29]. Jiyyo e-mitra clinic is a virtual PHC to expand outreach and primary health care by connecting local practitioners and health care workers virtually visiting patients along with a physician based in cities through video-conferencing technologies. A Chandigarh-based start-up called Jiyyo has developed this e-mitra clinic, which is a social-business model. The e-mitra clinic concept has been scaled up to 20 centres within Uttar Pradesh with private and public sponsored organizations. The consultation charges were kept to a minimum based on a not-for-profit, no loss model.

**mHealth application for hypertension prevention.** An mHealth application to raise awareness about hypertension and knowledge about its prevention in the general community was developed and tested in urban slums and rural pockets of Karnataka [31]. The app was conceptualised and developed by a multi-disciplinary team with expertise in arts, science, technology, business, nursing, medicine and design from the United States and India. The app incorporated a 3D interactive health education animation video and made in a culturally appropriate manner. The content of the video was related to prevention, diagnosis and management of hypertension. The app served as a digital literacy tool with options to collect data.

**Collaborative tele-psychiatry for the homeless mentally ill.** Tele-psychiatry system was another digital health innovation developed to support the homeless and mentally ill in Bengaluru Karnataka [32]. However, the important aspect of this innovation is human participation in the form of experts, led by PCP, collaboratively working to meet the rehabilitation needs of people with mental illnesses. This is another hub and spoke model very similar to the one described in Bihar for drug addiction service provision. The hub is the telemedicine centre within the department of psychiatry in NIMHANS with an expert consulting psychiatrist. The spoke is an urban PHC with the PCP and the patient. The expert provided guidance to the physician for screening, diagnosis and management of the patients with mental illnesses. The consultations lasted for about 15 minutes for assessment and 10 minutes for follow-up.

**ENTraview—Shruti Tele-otology program.** This digital technology was developed to screen hearing impairment in the community of 12 Indian states [31]. This study was implemented in 15 states in India namely Punjab, Haryana, Uttarakhand, Delhi, Uttar Pradesh, Bihar, West Bengal, Telangana, Assam, Karnataka, Tamil Nadu, Madhya Pradesh, Rajasthan, Gujarat, and Maharashtra. However, they did not report which 12 states from the 15 were included for the implementation of this project. The device was called ENTraview was developed by Medtronic, Inc. It works on a camera-enabled android device integrated with an otoscope, audiometric screening app, and a rechargeable battery-operated light source. ENTraview utilizes the digital camera in the smartphone to capture images of the tympanic membrane and stores it on the device. It also has a noise-isolating headset which enables audiometric screening in a semi-noisy environment. This device performs hearing screening, diagnosis, treatment and rehabilitation at the community level.

## Methodological quality of studies included in the review

We assessed the methodological quality of studies using the Mixed Methods Appraisal Tool (MMAT) [34]. Studies included in the review either assessed the DHI for its feasibility [28,29,31,32] or evaluated the effectiveness of the intervention using a pre-post quasi experimental design [27,30]. Only one study assessed the effectiveness of the DHI with a standard community-based comparison group [33]. The quality of this study was better compared to the other studies as appraised by the MMAT tool. More details of the quality of included studies are provided in the Table 3.

## Discussion

This review identified seven different kinds of DHIs that aimed in improving access to primary care in India. Some innovative and insightful aspects were noted from the above DHIs. The first and foremost being the conceptualisation of accessibility. Studies included defined accessibility from the perspective of availability of primary care services, the distance and geographical location of the health care facility. Hard to reach, remote, rural locations were an important geographical driver for the development of these DHIs. However, accessibility to healthcare needs to be reviewed from the perspective of not just availability but also affordability, approachability, appropriateness, and acceptability [35]. Additionally, attitudinal barriers for access to primary care need to be considered while conceptualising further AT innovations.

The next is the aspect of vulnerability. The studies in this review have included populations living in remote, rural hard to reach regions, and those homeless, poorly educated as a vulnerable group of people. However, other important aspects of vulnerability such as age (adolescence and elderly), gender (LGBTs), social situation (deprived, stigmatised), economic status (poverty), functional capacity (disability) were not considered during the conceptualisation and development of the DHI [36]. This is expected to create a huge gap and burden that is to

**Table 3. Critical appraisal of included studies using the mixed methods appraisal tool.**

|  | Item | Kulkarni 2020 | Gupta 2020 | Garner 2020 | Angrish 2020 | Prinja 2017 | Sagi 2018 | Choudhury 2021 |
|---|---|---|---|---|---|---|---|---|
| S1 | Clear research questions? | Yes | Yes | Yes | Yes | Yes | Yes | Yes |
| S2 | Data addresses the research questions? | Yes | Yes | Yes | Yes | Yes | Yes | Yes |
| 1 | Qualitative studies | NA | | | | | | |
| 2 | Quantitative studies RCTs | NA | | | | | | |
| 3 | Quantitative Non-randomised studies | | | | | | | |
| 3.1 | Participants representativeness | | | Yes | | Yes | | Yes |
| 3.2 | Are the measurements appropriate? | | | Yes | | Yes | | Yes |
| 3.3 | Complete of data? | | | Yes | | Yes | | C |
| 3.4 | Confounders accounted in the design and analysis? | | | No | | C | | Yes |
| 3.5 | Intervention administered as intended? | | | Yes | | No | | Yes |
| | Quantitative Descriptive studies | | | | | | | |
| 4.1 | Relevant sampling strategy | Yes | Yes | | C | | Yes | |
| 4.2 | Participants' representativeness | Yes | Yes | | Yes | | Yes | |
| 4.3 | Are the measurements appropriate? | Yes | Yes | | Yes | | Yes | |
| 4.4 | Risk of non-response bias low? | Yes | Yes | | C | | C | |
| 4.5 | Appropriate statistical analysis | C | Yes | | C | | C | |
| 5 | Mixed Methods Design | NA | NA | NA | NA | NA | NA | NA |

Y: Yes N: No C: Can't tell NA: Not applicable.

be met by further new or ratified innovations and initiatives from government, non-government, private and civil societies in India. Available DHIs to improve access have missed other key vulnerable groups that struggle to gain access to primary care. Although there is an exponential increase in DHI innovations for primary care worldwide, these interventions need to be inclusive both in terms of its design as well as its effects to bridge inaccessibility especially in the Indian context.

The last and most important aspect to be highlighted is the DHI per se. None of the included studies report about how it was developed. Although the focus was related to addressing the issues of access from a technological perspective, heavily depending on the use of smartphone and internet connectivity and access among the consumers of care, there was no information about the rationale for such a design and the evidence for the content and its quality from included studies. There is an immense need to develop digital interventions or digital health innovations systematically, that are also reproducible widely allowing the scale-up of their application universally [37]. A logical rationale, an evidence-based content and a systematic evaluation of these digital health interventions could be an important strategy to bridge the gaps in accessibility to primary care services in India.

Engagement of service users is also an important component of the systematic development of DHIs. The conceptualisation and development of DHI must be inclusive [38]. Primary users of health care services and primary care need to be involved in conceptualising and developing these technological innovations [38]. Studies included in the review had conceptualised and developed interventions based on the perspectives of services providers only. Perhaps this could be one reason why there are not many systematic evaluations of these DHI innovations. This reflects on the approach that was taken by included studies to evaluate the feasibility and effectiveness of their innovation for improving access to primary care. Studies included evaluated their innovations using a pre-post or cross-sectional method rather than a large scale randomised controlled trial to measure its clinical and cost effectiveness using

appropriate outcomes. Every year, each state within India conceptualises and develops important innovations which does not reach the stage of evaluation and scale up [39]. Hence there is an immense need to evaluate the digital health innovations with large-sized, adequately powered randomised controlled trials.

This review has several implications for the future of optimising DHI to improve access to primary care in India. Inclusive, systematic conceptualisation, development and evaluation of these digital health innovations is of high public health importance, especially in a country like India where there the needs and demands exponentially increase for such innovations.

## Conclusion

Digital health interventions, either as a standalone tool or when incorporated within the primary health system, with support or through training of grassroot level health workers, is a positive and a much-needed development in Indian health care. Primarily if the implementation of DHIs improves access to primary care among vulnerable populations living in the diverse, heterogenous Indian settings. To re-imagine an inclusive, digitally empowered health system in India, it is crucial to conceptualise, holistically develop and rigorously evaluate public health interventions made more easily and effectively accessible using assistive technology that will harbinger the bridging of gaps with reference to access to primary care in India.

## Supporting information

**S1 Checklist. PRISMA ScR checklist.**
(DOCX)

**S1 Text. Search strategy.**
(DOCX)

## Acknowledgments

This article has been written as part of the research for the *Lancet* Citizens' Commission on Reimagining India's Health System. The views expressed are those of the author(s) and not necessarily those of the Lancet Citizens' Commission or its partners.

## Author Contributions

**Conceptualization:** Lenny Vasanthan, Sindhu Kulandaipalayam Natarajan, Andrew Babu, Mohan S. Kamath, Sureshkumar Kamalakannan.

**Data curation:** Lenny Vasanthan, Sindhu Kulandaipalayam Natarajan, Sureshkumar Kamalakannan.

**Formal analysis:** Sindhu Kulandaipalayam Natarajan, Sureshkumar Kamalakannan.

**Funding acquisition:** Mohan S. Kamath, Sureshkumar Kamalakannan.

**Investigation:** Lenny Vasanthan, Sureshkumar Kamalakannan.

**Methodology:** Lenny Vasanthan, Sindhu Kulandaipalayam Natarajan, Andrew Babu, Mohan S. Kamath, Sureshkumar Kamalakannan.

**Project administration:** Lenny Vasanthan.

**Resources:** Sureshkumar Kamalakannan.

**Supervision:** Lenny Vasanthan, Sindhu Kulandaipalayam Natarajan, Sureshkumar Kamalakannan.

**Validation:** Sindhu Kulandaipalayam Natarajan, Andrew Babu, Mohan S. Kamath, Sureshkumar Kamalakannan.

**Visualization:** Lenny Vasanthan.

**Writing – original draft:** Lenny Vasanthan, Sureshkumar Kamalakannan.

**Writing – review & editing:** Lenny Vasanthan, Sureshkumar Kamalakannan.

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
