## [Decision Letter · Decision Letter 0]

27 Dec 2023

PGPH-D-23-02145

Assistive technology for improving access to primary care in India: a scoping review.

Dear Dr. Sureshkumar Kamalakannan

Thank you for submitting your manuscript to PLOS Global Public Health. After careful consideration, we feel that it has merit but does not fully meet PLOS Global Public Health’s publication criteria as it currently stands. Therefore, we invite you to submit a revised version of the manuscript that addresses the points raised during the review process.

EDITOR:

As per the initial feedback received kindly recheck the following-

1. The references seem to be mixed up and need correction.

2. The use of the term 'assistive technology' for digital interventions in PHC is not common- Would you like to rephrase it to what is understood better in global health eg- 'digital health interventions' for access to PHC in vulnerable populations or something like that?

3. Suggest to add / elaborate on what is unique about this scoping review, and why is it needed.

4. Kindly consider a deeper analysis and critical appraisal of the 7 case studies

5. NFHS 5 data is available now. Please include them in the introduction.

6. Methods

It is good that the protocol of this coping review was registered with the Open Science Framework (OSF). Kindly provide the registration number.

“The search was first run on the 29th of November 2022 and was an updated search was run six months later on 30 April 2023”. It is actually a gap of 5 months!

7. Utilization of Rayyan software for screening is mentioned only in the abstract. Please describe the details in the main article under methods section.

8. Was calibration exercise done at the step of selecting studies? If yes, how? and what did you find? Was inclusion criteria changed after the exercise?

9. A large number of studies were excluded. What are the reasons for exclusion? Please give details.

10. As per the definition, assistive technologies include any device, product or service. But no studies about services are included in the review?

11. Results

7 studies don’t represent all the regions of India? Provide the list of 12 Indian states in ENTraview - Shruti Tele otology program.

12. Conclusion

Please avoid writing only recommendations under conclusion. Rather it should answer the research question “Can assistive technologies/ Digital health interventions for the vulnerable population improve access to primary care in India?”

We look forward to receiving your revised manuscript.

Kind regards,

Manish Barman, MD., MSc., FRCP

Academic Editor

Journal Requirements:

1. Please provide separate figure files in .tif or .eps format only and remove any figures embedded in your manuscript file. Please also ensure all files are under our size limit of 10MB.

Additional Editor Comments (if provided):

Dear Authors

As per the initial feedback received kindly recheck the following-

1. The references seem to be mixed up and need correction.

2. The use of the term 'assistive technology' for digital interventions in PHC is not common- Would you like to rephrase it to what is understood better in global health eg- 'digital health interventions' for access to PHC in vulnerable populations or something like that?

3. Suggest to add / elaborate on what is unique about this scoping review, and why is it needed.

4. Kindly consider a deeper analysis and critical appraisal of the 7 case studies

5. NFHS 5 data is available now. Please include them in the introduction.

6. Methods

It is good that the protocol of this coping review was registered with the Open Science Framework (OSF). Kindly provide the registration number.

“The search was first run on the 29th of November 2022 and was an updated search was run six months later on 30 April 2023”. It is actually a gap of 5 months!

7. Utilization of Rayyan software for screening is mentioned only in the abstract. Please describe the details in the main article under methods section.

8. Was calibration exercise done at the step of selecting studies? If yes, how? and what did you find? Was inclusion criteria changed after the exercise?

9. A large number of studies were excluded. What are the reasons for exclusion? Please give details.

10. As per the definition, assistive technologies include any device, product or service. But no studies about services are included in the review?

11. Results

7 studies don’t represent all the regions of India? Provide the list of 12 Indian states in ENTraview - Shruti Tele otology program.

12. Conclusion

Please avoid writing only recommendations under conclusion. Rather it should answer the research question “Can assistive technologies/ Digital health interventions for the vulnerable population improve access to primary care in India?”

Looking forward to your revised submission

Thanks

Manish Barman

Reviewers' comments:

Reviewer's Responses to Questions

**Comments to the Author**

1. Does this manuscript meet PLOS Global Public Health’s publication criteria? Is the manuscript technically sound, and do the data support the conclusions? The manuscript must describe methodologically and ethically rigorous research with conclusions that are appropriately drawn based on the data presented.

Reviewer #1: Yes

Reviewer #2: Partly

2. Has the statistical analysis been performed appropriately and rigorously?

Reviewer #1: Yes

Reviewer #2: N/A

3. Have the authors made all data underlying the findings in their manuscript fully available (please refer to the Data Availability Statement at the start of the manuscript PDF file)?

Reviewer #1: Yes

Reviewer #2: No

4. Is the manuscript presented in an intelligible fashion and written in standard English?

Reviewer #1: Yes

Reviewer #2: Yes

5. Review Comments to the Author

Reviewer #1: The manuscript (Assistive technology for improving access to primary care in India) tries to cover the problem of assistive technology for improving access to primary care in India.

The main points that required amendment are as follows:

1. Some suggestions are required in recommendation part based on your findings

Reviewer #2: Overall, it is a good manuscript. I appreciate the efforts by the authors. However there are some queries/comments as mentioned below:

Introduction

Introduction is well written.

NFHS 5 data is available now. Please include them in the introduction.

Methods

It is good that the protocol of this coping review was registered with the Open Science Framework (OSF). Kindly provide the registration number.

“The search was first run on the 29th of November 2022 and was an updated search was run six months later on 30 April 2023”. It is actually a gap of 5 months!

Utilization of Rayyan software for screening is mentioned only in the abstract. Please describe the details in the main article under methods section.

Was calibration exercise done at the step of selecting studies? 

---

## [Decision Letter · Decision Letter 1]

29 Feb 2024

Digital Health Interventions  for improving access to primary care in India: a scoping review.

PGPH-D-23-02145R1

Dear Dr Kamalakannan,

We are pleased to inform you that your manuscript 'Digital Health Interventions  for improving access to primary care in India: a scoping review.' has been provisionally accepted for publication in PLOS Global Public Health.

Best regards,

Manish Barman, MD., MSc., FRCP

Academic Editor

Dear Authors

Thank you for addressing the reviewers suggestions

The manuscript can be recommended now for publication.

Reviewer Comments (if any, and for reference):

Reviewer's Responses to Questions

**Comments to the Author**

1. If the authors have adequately addressed your comments raised in a previous round of review and you feel that this manuscript is now acceptable for publication, you may indicate that here to bypass the “Comments to the Author” section, enter your conflict of interest statement in the “Confidential to Editor” section, and submit your "Accept" recommendation.

Reviewer #1: All comments have been addressed

Reviewer #2: All comments have been addressed

2. Does this manuscript meet PLOS Global Public Health’s publication criteria? Is the manuscript technically sound, and do the data support the conclusions? The manuscript must describe methodologically and ethically rigorous research with conclusions that are appropriately drawn based on the data presented.

Reviewer #1: Yes

Reviewer #2: Yes

3. Has the statistical analysis been performed appropriately and rigorously?

Reviewer #1: Yes

Reviewer #2: N/A

4. Have the authors made all data underlying the findings in their manuscript fully available (please refer to the Data Availability Statement at the start of the manuscript PDF file)?

Reviewer #1: Yes

Reviewer #2: Yes

5. Is the manuscript presented in an intelligible fashion and written in standard English?

Reviewer #1: Yes

Reviewer #2: Yes

6. Review Comments to the Author

Reviewer #1: None

Reviewer #2: Thanks for revising as per our comments and suggestions.

7. PLOS authors have the option to publish the peer review history of their article (what does this mean?). If published, this will include your full peer review and any attached files.

**Do you want your identity to be public for this peer review?** For information about this choice, including consent withdrawal, please see our Privacy Policy.

Reviewer #1: **Yes: **Muluken Tessema

Reviewer #2: No
